

# Analysis of Different Gray Zone Treatments in WRF-LES Real Case Simulations

Paula Doubrawa[1], Alex Montornès[2], Rebecca J Barthelmie[3], Sara C Pryor[4], and Pau Casso[2]

[1]National Renewable Energy Laboratory, Golden, CO, USA
[2]VORTEX, Barcelona, Spain
[3]Sibley School of Mechanical and Aerospace Engineering, Cornell University, Ithaca, NY, USA
[4]Earth and Atmospheric Sciences, Cornell University, Ithaca, NY, USA

*Correspondence to:* Paula Doubrawa (Paula.Doubrawa@nrel.gov)

**Abstract.** When conducting meso-micro scale coupled atmospheric simulations, it is crucial to ensure an adequate treatment of gray zone or terra incognita resolutions in which a large portion of the kinetic energy is naturally produced by the momentum balance equations in the model, while the remaining part still needs to be parameterized. In this work, we conduct three multi-day, real case, full-physics atmospheric simulations that are fully coupled from the meso to the micro scale and in which the

only difference is the treatment of boundary layer physics at the gray zone domain. One simulation uses a well-established parameterization, another uses its scale-aware version previously modified to accommodate gray zone resolutions, and a final one uses no parameterization at all and assumes that the gray zone domain can be run in large-eddy simulation (LES) mode. The simulated fields are cross-compared, and further compared to measurements collected during the Prince Edward Island Wind Energy Experiment. Use of LES in the gray zone domain influences the flow fields in a manner that is robust to temporal

averaging. The best predictions of vertical wind shear were found for the simulations in which the gray zone is parameterized, and the inclusion of a micro scale nest run in LES mode within the gray zone domains increased the model errors by producing overly homogeneous flow fields. The parameterized simulations also produced better agreement in terms of kinetic energy spectra at the two innermost simulation domains. In the gray zone domain, the energy decays as $f^{-3}$ throughout most of the spectral range considered. In the micro scale domain, the same is only seen in the low-frequency end of the gray zone spectral

range. In the high-frequency end, the energy decay follows a $f^{-1}$ slope. Outside the gray zone spectral range, the micro scale simulated spectra follow the expected $f^{-5/3}$ slope and produce good agreement with measurements.

## 1 Introduction

Recent advances in computational resources and atmospheric models have been driving the wind energy research community away from the Reynolds-Averaged Navier Stokes (RANS) approach, which does not permit a study of turbulence at the high

Reynolds numbers that characterize flow within wind farms. Instead, research is increasingly based on Large-Eddy Simulations (LES) in which the horizontal grid size ($\Delta_{xy}$) of the mesh is reduced to a point where Navier Stokes can be assumed to resolve all of the turbulence scales relevant to the problem at hand. However, due to their relatively high computational cost compared to RANS-based codes, LES have traditionally been used to examine ideal cases where a large number of assumptions simplify the



problem sufficiently for computational tractability and physical understanding (e.g., Schalkwijk et al. (2015); Shin and Hong (2013)). In this work, an important distinction is made between "idealized" and "real case" simulations based on the data being used as initial and boundary conditions. In idealized simulations, the model is often driven towards specified values of wind speed and temperature which are defined with the intent of representing canonical atmospheric conditions (e.g., Churchfield

et al. (2012)). In real case simulations, initial and boundary conditions are specified to match a past event of interest and often a reanalysis data set is used to drive the model.

Another important distinction relevant for this work is that between "full-physics" and "fluid dynamics" models. In addition to focusing on idealized conditions, LES studies for wind farm aerodynamics have historically used fluid dynamics solvers that do not include the full range of atmospheric physics and dynamics that impact flow within wind farms (e.g., cloud physics and

radiative processes). Thus far, full-physics real case LES focusing on atmospheric phenomena at micro scales have typically employed models that run un-coupled from the meso and macro scales (e.g., Kunz et al. (2000)) and for a time period on the order of hours (e.g., Liu et al. (2012)). For the purpose of this work, we distinguish between macro, meso, and micro atmospheric scales based on the duration ($T$) and spatial dimensions ($L$) of the most energetic processes within them as given in Table 1.

**Table 1.** Scales of atmospheric motions as defined for the present work based on duration and spatial dimension of most energetic processes within them.

|  | Duration | Spatial Dimension |
|---|---|---|
| Macro Scale | 1 year $< T <$ 1 day | $10^4$ km $< L < 10^1$ km |
| Meso Scale | 1 day $< T <$ 1 hour | $10^1$ km $< L < 10^0$ km |
| Micro Scale | $T <$ 1 hour | $L <$ 1 km |

Ongoing computational developments have only recently started to allow for full-physics, real case atmospheric simulations to be performed for wind energy applications. This can be achieved by coupling meso and micro scale models, and is a topic of active research for which two main approaches are being considered. One alternative is to perform meso scale simulations with an atmospheric model uncoupled from the micro scale and to subsequently use the meso scale flow to prescribe initial and boundary conditions to a secondary code that is then run on a smaller domain, a finer mesh, and in LES mode. Another

alternative is to fully couple the meso and micro scales within a single code. As a consequence of these new developments, two important questions have emerged. Firstly, how best to prescribe initial and boundary conditions to the LES and whether to allow feedback between the scales when the coupling is performed within a single code. Secondly, how best to treat the transition from meso to micro scales within numerical models (Sanz Rodrigo et al., 2016). The concept of "gray zone" (GZ) resolutions or "terra incognita" was coined to describe the spatial scales $\mathcal{O}(10^2 - 10^3)$ m at which Navier Stokes is able to

resolve a substantial fraction of the kinetic energy in the atmospheric boundary layer (ABL), while still needing to model the remaining part with physical parameterizations (Wyngaard, 2004). Under the first coupling approach, the GZ is avoided but turbulence generation methods must still be applied to the meso scale fields before they can be used as driving inflow to LES





(Haupt et al., 2015; Mirocha et al., 2013). Under the second coupling approach, an adequate treatment of GZ resolutions is required and may be achieved by developing more flexible parameterizations (Shin and Dudhia, 2016; Ito et al., 2015) or by avoiding the GZ domains altogether and relying on turbulence generation strategies alone to couple meso and micro scale domains (Muñoz-Esparza et al., 2017).

A lot of the recent work in this field has focused on meso-micro coupling within the Weather Research and Forecasting (WRF) model (Skamarock et al., 2008), which is a widely used framework for idealized and real case meso scale atmospheric simulations that can be run in LES mode by reducing $\Delta_{xy}$ down to the micro scale and disabling several parameterizations. For example, Shin and Hong (2015) and Ito et al. (2015) used idealized WRF-LES as benchmarks and developed scale-aware capabilities within existing ABL parameterizations (ABLPs) to regulate their role at GZ resolutions. So far, these and other

ABLPs expanded to accommodate GZ resolutions are limited to convective boundary layers (CBLs) and have mostly been verified against reference idealized LES (e.g., Shin and Dudhia (2016); Ito et al. (2015); Kitamura (2015); Honnert et al. (2011)). The only existing work evaluating meso-micro coupled simulations with WRF for a real case avoided the GZ by using a high nesting ratio and going directly from the meso (1 km) to the micro scale (90 m) domain (Muñoz-Esparza et al., 2017). While this approach provided satisfactory results when compared with observations over a diurnal cycle, it is still important to

continue the development of techniques to accurately simulate atmospheric flow at intermediate resolutions.

Very little work has been done to understand how the simulated flow responds to different treatments of GZ resolutions under real case LES that are fully coupled to the meso scale. Boutle et al. (2014) applied aircraft observations to evaluate simulations of stratocumulus formation under the proposed scale-aware modifications to the Met Office Unified Model, while nesting domains from 4 km to 100 m and considering a period of 2 days. They found that the performance of the scale-aware

ABLP at the GZ matched the performance of the well-established one-dimensional large-scale ABLP used for coarser grids, and that of the three-dimensional small-scale parameterization typically used for micro scale grid sizes. Another study by Shin and Hong (2015) proposed a scale-aware ABLP in WRF and validated it for real cases considering 24 hours of observational data focusing on the development of a convective roll at $\Delta_{xy} = 333$ m. They found that the newly proposed scheme enhanced the simulation of vertical motions under convective conditions but highlighted the need for further improvements which will

cover a wider range of simulation scenarios. A different study was carried out by Schalkwijk et al. (2015), where the effect of modifications to the GALES model was assessed by comparing one year of simulation data at $\Delta_{xy} = 100$ m to observations from a meteorological mast. They found that the best agreement of simulations with observations occurred for ABL parameters that are explicitly resolved (i.e., instead of parameterized) thus further highlighting the need for long-term real case LES and for further research in meso-micro scale model coupling.

More research is needed to guide future developments of scale-aware one-dimensional ABLPs that can reach down from the meso to the micro scale, and of three-dimensional ABLPs that can reach up from the micro to the meso scale. The research presented herein addresses this need by evaluating different approaches for treating GZ resolutions in full-physics real case LES of the atmosphere that are fully coupled to the meso scale. This is the first study in which real case WRF-LES simulations are performed for a period longer than one day. Moreover, it is the first such study that focuses on quantifying the differences

in simulated flow arising from different GZ treatments. We focus on flow parameters of relevance to wind engineering and



quantify differences between three simulations in which the GZ is treated differently by being run with a well-established ABLP, its scale-aware version, and no ABLP at all. The simulation output is compared to observational data collected during the Prince Edward Island Wind Energy Experiment (Barthelmie et al., 2016). This specific location was selected for our analysis because the terrain complexity and roughness changes found at the site warrant the use of LES in contrast to coarser,

meso scale simulations which are unable to resolve the complex flow characteristics at the site. The analysis considers a 16-day period thus enabling an assessment of the simulations performance under a range of atmospheric conditions and several diurnal cycles. The methodology is discussed in Section 2. A cross-comparison of the model simulations is given in Section 3, and a comparison between the simulations and measurements in Section 4. A final discussion can be found in Section 5.

## 2  Methodology

### 2.1  Boundary Layer Treatment at the Gray Zone

Parameterized phenomena in meso scale atmospheric models (such as WRF, when it is not run in LES mode) include radiative transfer, convection, and ABL physics. In ABLPs, a distinction is made between local vertical transport between adjacent grid levels and non-local vertical transport via strong updrafts that span the ABL depth (Frech and Mahrt, 1995). Non-local ABL schemes such as the well-established YSU (Hong et al., 2006) include a non-local term in addition to the local transport. The

SH scheme (Shin and Hong, 2015) is an expansion of YSU in which the amount of SGS vertical transport that needs to be parameterized is regulated based on $\Delta_{xy}$ and on the strength of the non-local transport itself.

All simulations presented herein use the YSU ABLP for domains with a grid resolution coarser than the GZ resolution, and are run in LES mode for the innermost domain in which the grid resolution is finer than at the GZ. The WRF source code was modified to enable meso-micro scale coupling based on the potential temperature perturbation method of Muñoz-Esparza

et al. (2014, 2015). Previous work has shown that this modification results in an improvement to simulations of wind speed and turbulence intensity under different meteorological regimes and terrain complexities (Montornès et al., 2016b, a).

Hereinafter, the three simulations conducted are referred to as YSU_LES, SH_LES, and LES_LES (Table 2) based on how they handle the ABL physics at the GZ and at the innermost domain. In the LES_LES simulation, the ABLP is switched off in the GZ domain and the SGS energy is modeled with a local closure in which a 1.5-order prognostic equation for turbulent

kinetic energy (TKE) is used. This approach is recommended when the energy-containing eddies are substantially larger than the grid resolution (Shin and Hong, 2015) which is not necessarily the case at the GZ. Therefore the GZ treatment in our LES_LES simulation is not necessarily a physical choice, but can be justified by the existence of a nested domain which is appropriately run in LES mode.





**Table 2.** Treatment of atmospheric boundary layer physics at each simulation domain for the three simulations conducted in the experiment.

| Simulation Name | Atmospheric Boundary Layer Parameterization (ABLP) | | |
| | Meso Scale Domains | Gray Zone Domain | Micro Scale Domain |
| | $\Delta_{xy} = 9\,\mathrm{km}, 3\,\mathrm{km}, 1\,\mathrm{km}$ | $\Delta_{xy} = 333\,\mathrm{m}$ | $\Delta_{xy} = 111\,\mathrm{m}$ |
|---|---|---|---|
| YSU_LES | YSU | YSU | None |
| SH_LES | YSU | SH | None |
| LES_LES | YSU | None | None |

Because the differences between YSU and SH pertain to the strength of the vertical transport, we expect differences in vertical flow velocities and in vertical fluxes of heat and momentum between these two simulations. In idealized simulations, the system complexity is lower and substantial differences between YSU_LES and SH_LES are only expected during convective conditions (i.e. strong non-local transport) while under neutral and stable stratification, SH should default to its underlying

YSU structure. On the other hand, the higher level of complexity in a real case simulation precludes the predetermination of the exact effects that SH will have on the results. This complexity further highlights the need for the present study in which these differences are investigated. Prior to the analysis, the only claim that can be made for the real case simulations conducted is that a distinction between YSU_LES and SH_LES is only expected during isolated periods throughout the simulation, when the differences between YSU and SH are noticeable. Instead, the largest difference is expected to be found between LES_LES

and the other two simulations in which the ABL is parameterized.

## 2.2   Study Domain and Measurements

The simulations were conducted for a domain centered on the North Cape of Prince Edward Island for the period of the Prince Edward Island Wind Energy Experiment field campaign (Barthelmie et al., 2016) at the Wind Energy Institute of Canada in May 2015. The narrow and flat island spans $\sim 2$ km across the measurement site (Fig. 1). To the west, a 10-14 m escarpment

marks the transition from ocean to continent. An 80 m meteorological mast compliant with the International Electrotechnical Commission standards and equipped with 3D Gill Windmaster Pro sonic anemometers collected 10 Hz wind and temperature measurements at the heights ($z$) 20, 40, and 60 m above ground. The mast location is $\sim 900$ m ($\sim 400$ m) from the coast in the north (west) direction.

The data from each sonic anemometer were subject to despiking, detrending and coordinate rotation when calculating vari-

ances and covariances. Data were further conditionally sampled to exclude wind directions associated with wind turbine wakes (unless otherwise noted in the analysis) and then used to compute streamwise, transverse, and vertical wind speeds, and vertical fluxes of heat and momentum. Conditions during the experiment period were dominated by onshore flow over the escarpment (SW-NW flow was observed 75% of hours) and the Obukhov length as computed from the sonic anemometers indicate unstable conditions on $\sim 10\%$ of hours, and stable conditions on $\sim 25-30\%$ of hours (Barthelmie et al., 2016).





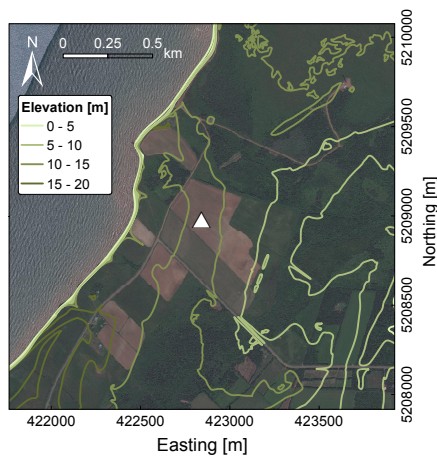

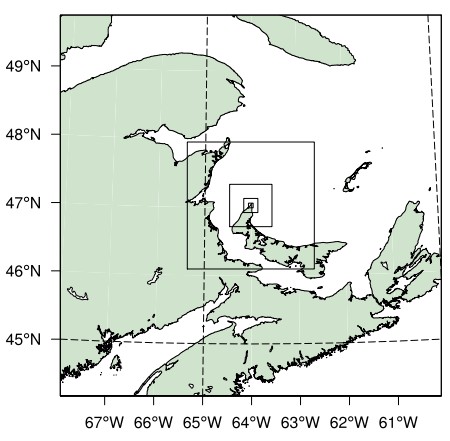

(a) Site satellite imagery and elevation.

(b) WRF simulation domains.

**Figure 1.** Measurement site terrain elevation, satellite imagery from Environmental Systems Research Institute, and location of meteorological mast (triangle) (a). Simulation domains from $\Delta_{xy} = 9$ km to $\Delta_{xy} = 111$ m (b).

## 2.3 Simulations

We performed three simulations using the Advanced Research WRF core version 3.7.1 that differ only in their treatment of the GZ domain as described in Section 2.1. A total of five domains (with 70 x 70 grid points) were configured, centered at the meteorological mast location following a set of telescopic nests from 9 km to 111 m (Fig. 1b). Feedback between the scales is

disabled so that the GZ fields can be analyzed without any influence from the turbulence that develops at the innermost domain which is run in LES mode across all simulations. Fifty vertical levels are defined throughout the domain, which are distributed every $\sim 10$ m in the ABL and stretched in the free atmosphere. Initial and boundary conditions for all domains are taken from the Climate Forecast System Reanalysis (Saha et al., 2010) at a spatial resolution of $\sim 38$ km. Topography for the highest resolution domain is taken from the Shuttle Radar Topography Mission (Jarvis et al., 2008) at a 90 m spatial resolution, and

land use from GlobCover (Bontemps et al., 2011) at a $\sim$300 m spatial resolution.

     The analysis of model output focuses on flow parameters in the ABL surface layer. The model was run for a 16-day period and time series were saved exclusively for the grid point closest to the meteorological mast location (Fig. 1a) for seven heights above ground between $\sim 10$ m and $\sim 180$ m. The model saves historic output with a temporal frequency of 4 Hz, which are immediately used to compute 10 minute-averaged diagnostics.

## 2.4 Variables and notation

All analyses are performed for a fixed location (i.e., the meteorological mast) and consider vertical profiles of 10-minute mean quantities. Hereinafter, the overbar ($^{-}$) is used to denote 10-minute averages when referring to (co)variances, but is omitted for all other variables to simplify the notation. We focus on flow variables of relevance to wind energy such as horizontal wind





speed, wind direction, vertical wind shear, and turbulence intensity. Turbulence intensity is only calculated for time stamps in which the horizontal wind speed is $\geq 3$ m s$^{-1}$ which are most relevant to wind energy applications. While the focus of this work is on variables of interest to wind energy, we also choose to include vertical flow velocities and vertical fluxes of heat and momentum which are crucial to characterizing turbulence and atmospheric stability. Perturbations are calculated

by subtracting the 10-minute mean from the 4 Hz instantaneous values as the simulation output is produced, and then used to obtain turbulence diagnostics at a 10-minute frequency. The turbulence diagnostics computed are kinematic heat flux and turbulent kinetic energy $TKE = TKE_\Delta + TKE_{SGS}$. $TKE_\Delta$ represents the amount of TKE that is explicitly resolved by the model at a grid resolution $\Delta_{xy}$ including the ABLP budget, and $TKE_{SGS}$ is the amount that is produced by the turbulence closure model when no ABLP is used and the simulation is run in LES mode. The SGS partition is diagnosed by the model

directly, and the resolved partition is calculated as $TKE_\Delta = \frac{1}{2}\left(\overline{u'^2} + \overline{v'^2} + \overline{w'^2}\right)$.

Differences in simulation output that arise from varying the model configuration are discussed in Section 3 without reference to the measurements. To avoid selecting one of the three cases as the reference benchmark, these differences are quantified by a deviation from each simulation to the ensemble average of the three simulations. The model performance relative to observations is quantified in a similar fashion by subtracting the modeled fields from the measured values. To facilitate a later

comparison with measurements, the majority of the model cross-comparison section focuses on the measurement heights of 20, 40, and 60 m by linearly interpolating from the model levels to the target height.

## 2.5    Spectral Analysis

Spectral methods are used to compare the kinetic energy content of each data set in the frequency domain. Comparisons are presented between the three simulations, between the two innermost domains for each simulation, and between simulations and

measurements. The power spectral density (PSD) values are computed for time series of 10-minute mean wind speeds using the Welch method. Previous work considering onshore and offshore sites has found that wind speed measurements collected at a frequency of 10 minutes are sufficient to accurately estimate the PSD in the atmospheric surface layer (Larsén et al., 2016). Because the spectral analysis is based on relative comparisons between the data sets, it is not concerned with energy leakage at low frequencies that could be caused by non-stationarity in the time series, and a hanning window with a length of 1 day

is applied when computing PSD values. With a temporal frequency of 10 minutes and a window length of 1 day, the spectra are limited to 1 day$^{-1} < f < 20$ min$^{-1}$ which is sufficient when focusing on GZ phenomena. High frequency fluctuations are therefore not included in the spectra, but are instead contained in the analysis of turbulence intensity and TKE which are computed from the 4 Hz model output.

## 3    Cross-Comparison of Simulations

In this section the three simulations are cross-compared to quantify their sensitivity to different treatments of the ABL physics at the GZ scale. Section 3.1 focuses on horizontal and vertical wind speeds, Section 3.2 on vertical wind shear and turbulent kinetic energy, and Section 3.3 discusses the energy content of each simulation in the frequency domain.



### 3.1 Horizontal and Vertical Flow

To compare the flow predictions across the three simulations, an ensemble average was first computed and then differences between each simulation and this reference ensemble were obtained at each time step. The values given in Fig. 2 show the mean and standard deviation of these differences at each hour of the day, including 16 days of simulation and 6 10-minute values in each hour. On average, the LES_LES simulation produced higher values of horizontal wind speed $U$ and of vertical velocities $w$ than the parameterized simulations. For both wind speed and vertical flow, there is no clear tendency of the mean differences with time of day. Rather, a consistent difference is seen that decreases with height for $U$ and that remains of the same magnitude for $w$ across the three heights. As expected, differences between YSU_LES and SH_LES are intermittent and average out when a long time series is considered. The standard deviation of these differences is also substantially larger for LES_LES than for the other two simulations, indicating that running the GZ domain in LES mode produces more fluctuations in the solution than activating a traditional or scale-aware ABLP.

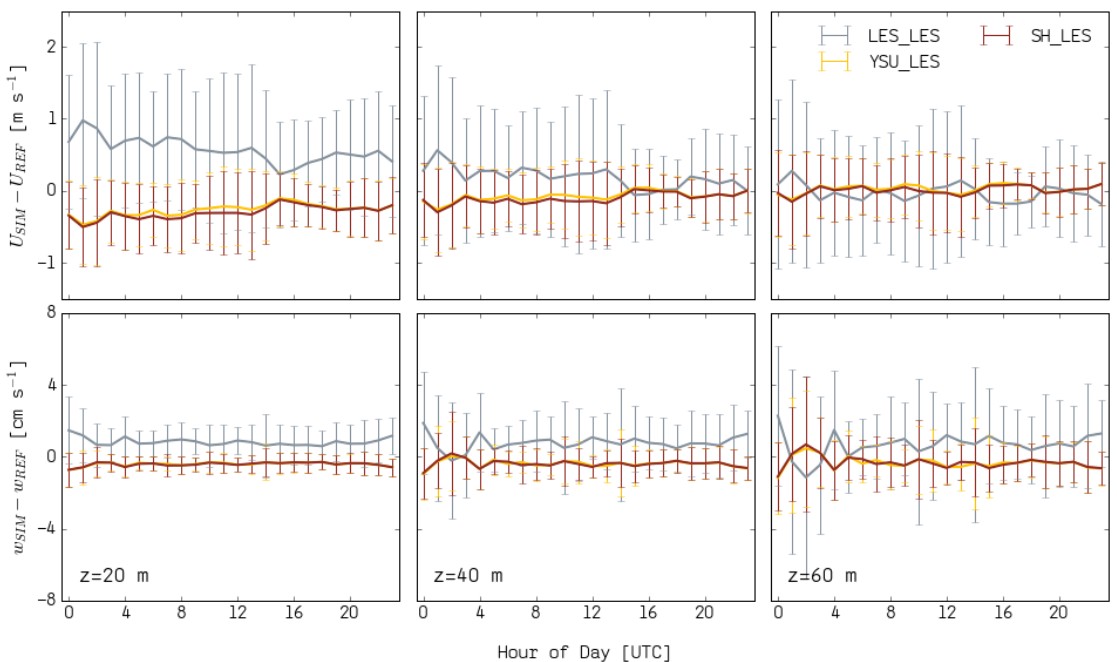

**Figure 2.** Differences between each simulation and their ensemble average for horizontal wind speed (top) and vertical flow velocity (bottom) for the heights 20 m (left), 40 m (middle) and 60 m (right). Error bars give the mean ± standard deviation of differences at each hour of day. Values are for the GZ domain.

When considering all model levels for which output was saved (Fig. 3), it is clear that LES_LES produces higher wind speed estimates only in the lower portion of the profile, and that the opposite is true above $\sim 60$ m. Mean differences across the simulations for vertical flow decrease with height, while the standard deviation of these differences increases for the




three simulations. Finally, a consistent difference is seen for wind direction, where `LES_LES` predicts flow that is veered (i.e., rotated in the clockwise direction) by $\sim 10°$ relative to `YSU_LES` and `SH_LES` throughout the vertical layer considered. These results suggest that running the GZ domain in LES mode has a large impact on the results, even when multi-day averages are considered. Mean horizontal wind speeds at some heights differ by more than 1 m s$^{-1}$ on average, with standard deviations

of the same magnitude which can be critical when using meso-micro coupled simulations for wind energy applications. The same is true for wind direction, with standard deviation values (as computed from the 10-minute periods) indicating that one-third of periods exhibit a difference in wind direction greater than 30°. These differences cannot be neglected when performing WRF-LES simulations for wind plant optimization via layout design or wake-redirection control strategies, which is a potential application of these simulations as a robust meso-micro coupling strategy is developed. A better understanding of the physical

reasons behind these differences is investigated in the next sections.

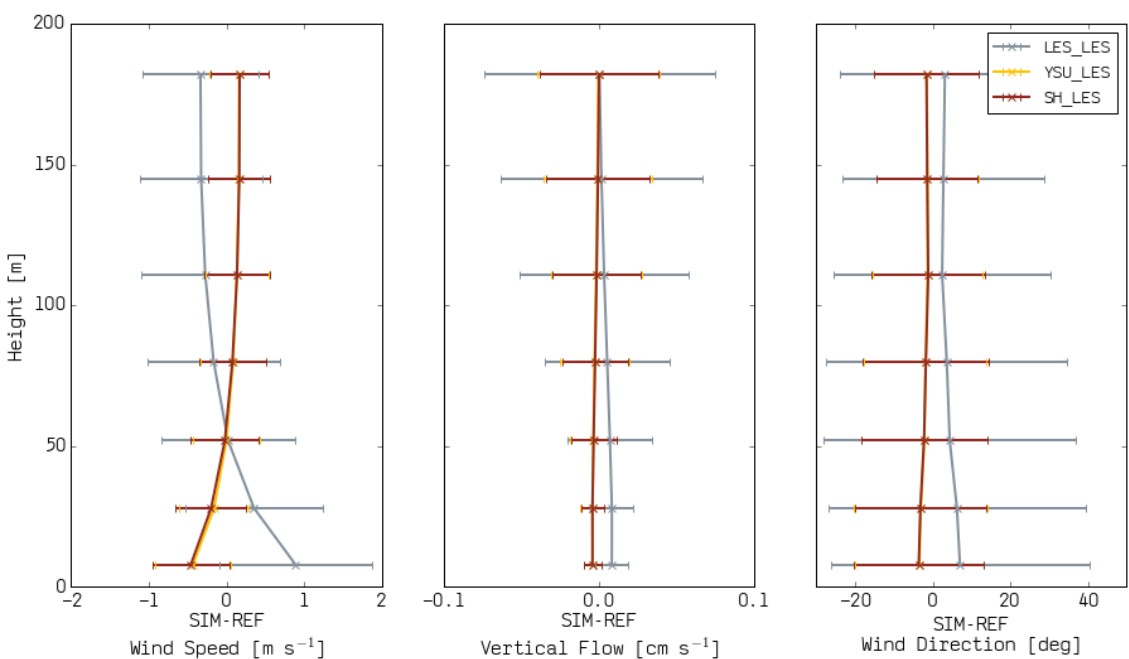

**Figure 3.** Vertical profile of differences between each simulation and the ensemble average for horizontal wind speed (left), vertical flow velocity (middle) and wind direction (right) for the GZ domain. Errorbars give mean $\pm$ standard deviation considering the entire time series.

### 3.2   Shear and Turbulence

The probability density function of 10-minute mean values of vertical wind shear across a nominal turbine rotor plane (extending between $\sim 30-110$ m) also differs substantially between `LES_LES` and the simulations in which the ABL is parameterized. At the GZ and micro scale resolutions `YSU_LES` and `SH_LES` are almost indistinguishable (Fig. 4). However, the difference

between these simulations and `LES_LES` is pronounced especially at the GZ domain where the shear distribution derived from `LES_LES` has a much lower mean and standard deviation, with all values below 0.03 s$^{-1}$. This is consistent with the results





discussed in Section 3.1 which indicate higher vertical velocities for the LES_LES simulation, resulting in more homogeneous flow and less vertical wind shear in the surface layer. This result is also evident in the micro scale domain, where the mean and standard deviation of shear are still lower for LES_LES than for the other two simulations, evidence of the importance of the ABL treatment at the GZ when conducting simulations with multiple nested domains.

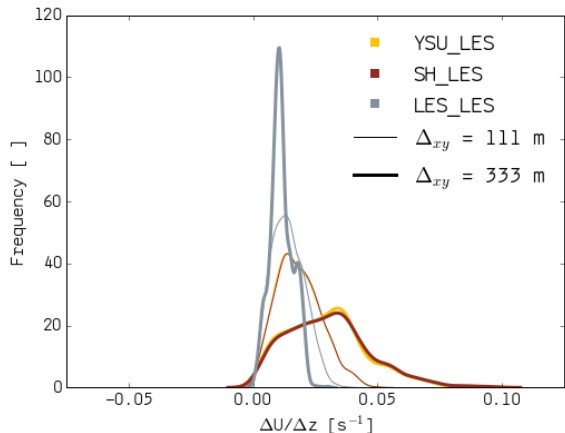

**Figure 4.** Curve-fitted histograms for vertical wind shear between $\sim 30 - 110$ m, considering the 16 days of simulation. Values shown for GZ and micro scale domains.

Fig. 5 shows the TKE budget from the three simulations for the highest measurement height ($z = 60$ m) and allows us to examine how the treatment of the ABL physics at the GZ resolution affects the total TKE budget within the two innermost domains. At the GZ domain, differences between LES_LES and the two other simulations are obvious, and result from the fact that LES_LES produces large amounts of subgrid TKE for wind speeds $> 8$ m s$^{-1}$, while the other two produce none. When $TKE_{SGS}$ is combined with $TKE_{\Delta}$ which is of the same magnitude for the three simulations, large differences result in the total TKE budget between LES_LES and the other two simulations. High TKE at $\Delta_{xy} = 333$ m in LES_LES contributes to explaining the low vertical wind shear values seen in this simulation (Fig. 4).

In the micro scale nest, differences in TKE between LES_LES and the other two simulations are lower in magnitude but still clearly noticeable. These differences are similar to those seen in the GZ domain, thus suggesting propagation down the model chain. In the micro scale domain, these differences end up modulating the budget of SGS, modeled TKE and leading to differences in $TKE_{SGS}$ across the different simulations. While SH_LES and YSU_LES present median $TKE_{SGS}$ values that increase with wind speed up to 15 m s$^{-1}$, LES_LES shows a local TKE maximum between 10 and 12 m s$^{-1}$ coinciding with the peak of subgrid TKE production in the GZ domain when run in LES mode. These results suggest that the subgrid TKE production at the GZ domain enabled by running it in LES mode inhibits the subgrid TKE production at the micro scale nest. In other words, the use of ABLPs at the GZ results in lower TKE at the GZ domain but in higher TKE at the innermost nest.





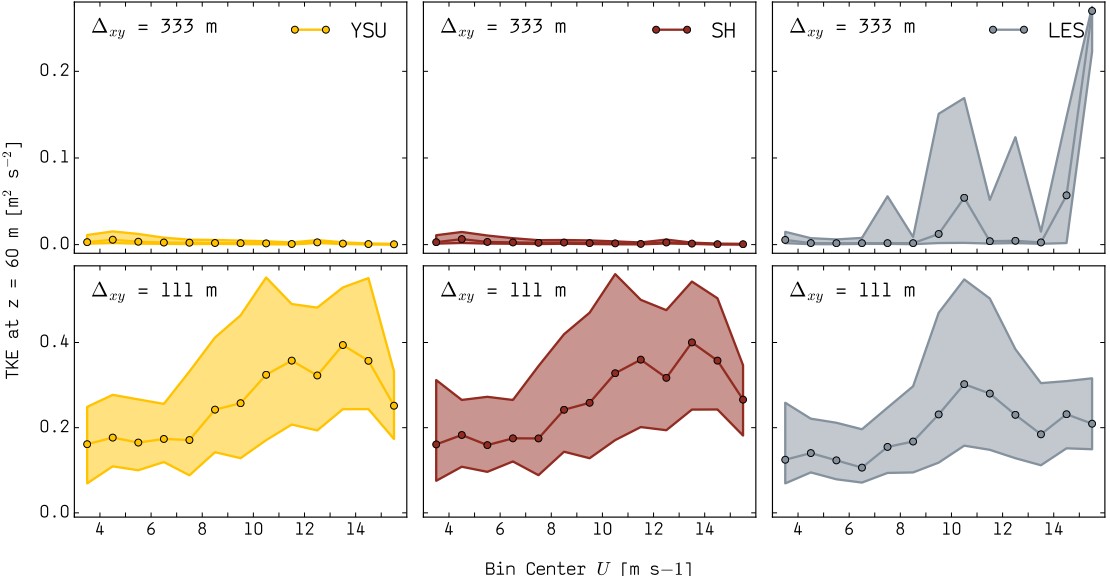

**Figure 5.** Median (markers) TKE at 60 m [m$^2$ s$^{-2}$] as a function of wind speed [m s$^{-1}$] discretized in 1 m s$^{-1}$ bins in the GZ (top) and micro scale (bottom) domains. Solid lines delimiting shaded area give $25^{th}$ and $75^{th}$ percentile values.

### 3.3 Energy Spectra

The results presented in this section are based on the spectral analysis methods described in Section 2.5. The PSD for wind speeds considering the entire 16-day time series are shown for the three simulations in Fig. 6. As expected, `YSU_LES` and `SH_LES` exhibit higher PSD values when compared to `LES_LES` because the ABLP (enabled in `YSU_LES` and `SH_LES`)

acts to generate turbulence at length scales higher than the grid size. Note that this analysis is different from the one presented in Section 3.2 which encodes turbulence information frequencies up to 4 Hz. An important result from this spectral analysis is that this difference between `LES_LES` and the other two simulations is seen throughout the spectrum, and is not concentrated in the spectral tail as might be expected.

Fig. 6 also indicates that the energy decay at the GZ domain follows a $f^{-3}$ slope instead of the $f^{-5/3}$ expected for meso scale

and boundary layer turbulence at frequencies larger than 1 day$^{-1}$ (Larsén et al., 2016). This result suggests that the GZ domain is lacking energy for the three simulations, and that a better treatment of turbulence at this scale is needed as will be further discussed in Section 4.3. At the micro scale domain, the spectra are substantially more energetic at higher frequencies and exhibit the $f^{-1}$ decay seen in Muñoz-Esparza et al. (2017) for these frequencies, also exclusively in the nested LES domain. Because simulation data are unavailable at temporal resolutions higher than 10 minutes, an in-depth analysis of the energy

content at higher frequencies is not possible and we cannot determine whether the expected spectral behavior is achieved for $f > 20$ min$^{-1}$.





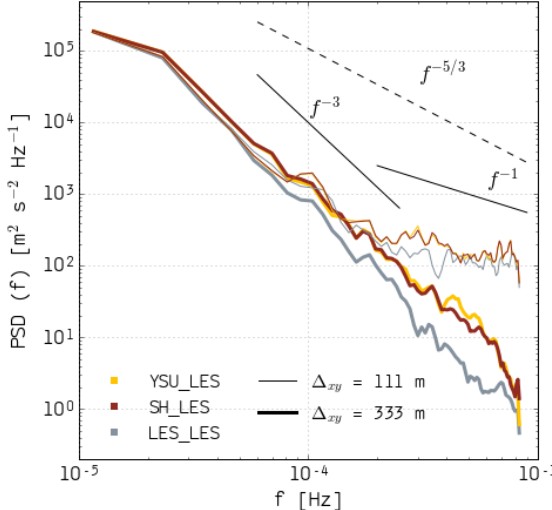

**Figure 6.** Power spectral density as a function of frequency for the 16-day time series of 10-minute mean horizontal wind speeds at $z = 60$ m for the three simulations at the GZ and micro scale domains.

## 4 Comparison of Simulations and Measurements

In this section the three simulations are compared to measurements from sonic anemometers to quantify how much improvement can be gained from having a micro scale domain nested within a multi-scale simulation framework, and to determine which GZ treatment produces the overall best results in terms of mean flow velocities and turbulence. Section 4.1 focuses on horizontal wind speed, Section 4.2 on vertical wind shear and turbulent kinetic energy, and Section 4.3 compares the energy content of each data set in the frequency domain. Thus far the simulations have been cross-compared without reference to the measurements to illustrate the sensitivity of flow parameters to different GZ treatments in each simulation. In this section, we evaluate the simulated flow fields relative to the observational data set described in Section 2.2. Whenever not explicitly specified, the comparison considers simulation data from the micro scale domain.

### 4.1 Horizontal Wind Speed

Fig. 7 shows the distribution and magnitude of horizontal wind speed errors as a function of different atmospheric conditions in terms of wind speed, wind direction, and turbulence intensity. An obvious and important result from this analysis is that the YSU_LES and SH_LES error distribution is almost indistinguishable regardless of the atmospheric conditions considered, indicating that the differences between these two simulations discussed in Section 3 are lower in magnitude than the model error. The data distribution histogram above the leftmost subplot shows that most of the observed wind speeds are between 2 and 14 m s$^{-1}$, and that wind speed errors exhibit a decreasing tendency with increasing wind speed. Although time stamps in which the meteorological mast was subjected to a wind turbine wake were filtered out, the model still over-estimates the wind





speeds below 10 m s$^{-1}$ with very wide error distributions between 2 and 6 m s$^{-1}$. Above 10 m s$^{-1}$, the model performed better for the three simulations.

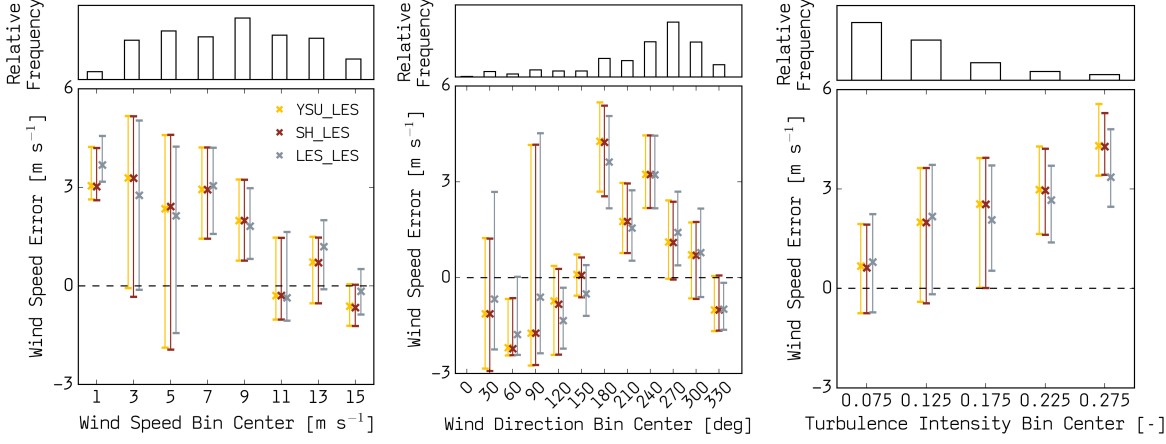

**Figure 7.** Wind speed errors [m s$^{-1}$] at $z = 60$ m averaged over different measured atmospheric conditions characterized at the same height: wind speed (left), wind direction (middle), and turbulence intensity (right). Markers represent the median error, and whiskers give $25^{th}$ and $75^{th}$ percentile values for each bin. Bars indicate how much of the total data considered belong in each bin.

There is a systematic pattern of wind speed error distribution as a function of wind direction, across the three simulations. During the few cases of winds between the northeastern and southeastern sectors ($wdir < 165°$, flow over the island) the
model mostly under-estimated the wind speed magnitude. Conversely, an over-estimation is seen for westerly and southerly winds ($165° \leq wdir < 315°$) which make up the majority of the data set and represent on and offshore flow (Fig. 1). In terms of differences between simulations, let us consider only the sectors with higher measurement density ($wdir > 165°$). We can see that `LES_LES` performed slightly better for southerly winds and the other two simulations for westerly winds. This result may be related to the onshore footprint at lower measurement heights (e.g., $z = 20$ m) at this site as discussed in Barthelmie
et al. (2016). This onshore footprint would result in the development of an internal boundary layer brought on by changes in roughness and orography, which may fail to be reproduced in a simulation where the ABL is not parameterized thus explaining the slightly higher mean error values for `LES_LES` (when compared to `YSU_LES` and `SH_LES`) for westerly flow.

In terms of turbulence intensity, a clear pattern can also be seen. Overall, the median wind speed error is higher for higher levels of turbulence intensity where wind speeds tend to be lower. Regarding the differences across simulations, the model
errors are larger for `LES_LES` at the low turbulence intensity bins ($TI < 0.15$) and lower for `LES_LES` for the bins $0.15 \leq TI < 0.25$. Note that the majority of the data lie within TI<0.15, leading to a temporal average of wind speed errors that is slightly higher for `LES_LES` than for `YSU_LES` and `SH_LES`.




## 4.2 Shear and Turbulence

Fig. 8 shows the distribution of shear across the measurement layer ($z = 20$ m to 60 m) during the entire period for which measurements are available after quality control. An important result from this analysis is that the best model performance is seen for the GZ domain when the ABL is parameterized. The simulation run in LES mode largely underestimates the observed shear across this layer, which is also true for the three simulations at the innermost domain ($\Delta_{xy} = 111$ m). This result indicates that the energy produced by the ABLP at the GZ is important to the simulation of vertical variations of wind speed in the surface layer, and that running a simulation in LES mode at the GZ compromises the model results in the presence of high shear (e.g., under stable atmospheric conditions which were commonly observed during the Prince Edward Island Wind Energy Experiment). These results also suggest that adding an extra domain with a higher horizontal resolution to the simulation with one-way coupling between the meso and micro scales (as is done here, with feedback being disabled between the domains) does not necessarily improve the simulated flow fields as might be expected. In other words, an adequate treatment of coarser domains is more important to obtaining accurate simulations of vertical wind shear than increasing the resolution of the grid by nesting the domains down to what can be assumed as an LES scale.

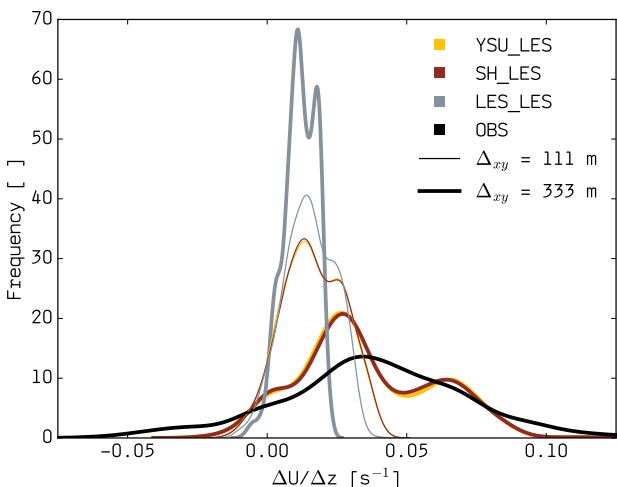

**Figure 8.** Histogram for vertical wind shear $\Delta U/\Delta z$ [s$^{-1}$] between 20 and 60 m, for all 10-minute periods for which coincident model output and observations are available.

As shown in Fig. 9 all simulations reproduce the approximate value of total TKE at low wind speeds, but fail to simulate the magnitude of the observed increase in TKE with wind speed. The wide range of TKE values measured at $z \geq 40$ m is underestimated by all simulations. The difference between simulations is much smaller than the difference between the simulations and measurements, but overall `YSU_LES` and `SH_LES` result in bin-wise median values that are slightly closer to the measurements than those produced by `LES_LES`.




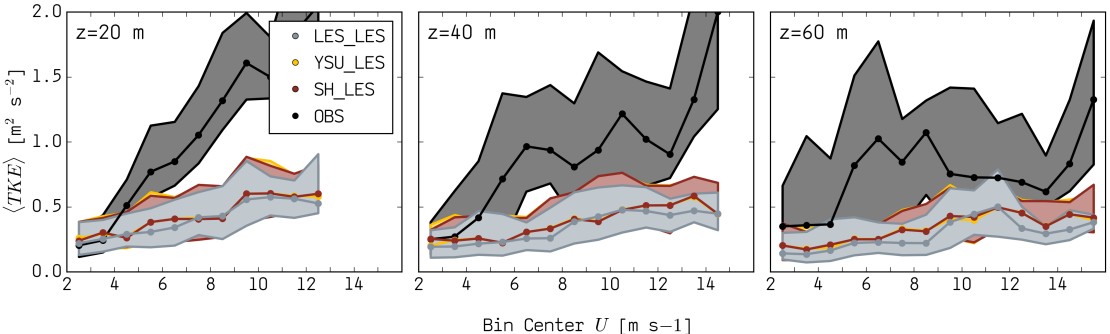

**Figure 9.** Median (markers) TKE [m$^2$ s$^{-2}$] as a function of wind speed (discretized in 2 m s$^{-1}$ bins) for observations and simulations at $z =$ 20 m (left), 40 m (middle), and 60 m (right). Vertical axes are identical. Solid lines and shading in between give $25^{th}$ and $75^{th}$ percentile values. Data included are observations (black), SH_LES (red), LES_LES (gray) and YSU_LES (yellow).

In terms of kinematic heat flux, different simulations perform best under different conditions. Regardless of the GZ treatment, the model does not reproduce the mean diurnal cycle of heat flux especially at $z = 20$ m (Fig. 10), indicating that this could be a limitation of the surface layer physics treatment in the model (i.e. via land surface and surface layer parameterizations?) and not necessarily of the ABL treatment, which is beyond the scope of the present study. The agreement between simulations and
5  measurements improves above 20 m, where the measured flux is lower in magnitude and the model produces higher estimates. The diurnal cycle that seems to appear in the model data starting at $z = 40$ m captures the magnitude of negative and positive peaks in the observations but these peaks are shifted by $\sim 6$ hours from the observed values. At $z = 60$ m there are discernible differences between the YSU_LES and SH_LES simulations both for negative and positive flux periods. Namely, in SH_LES the magnitude of the mean flux is slightly lower than in YSU_LES, because the scale-awareness of the ABLP is modulating
10  the effect of the physical scheme on the fluxes.

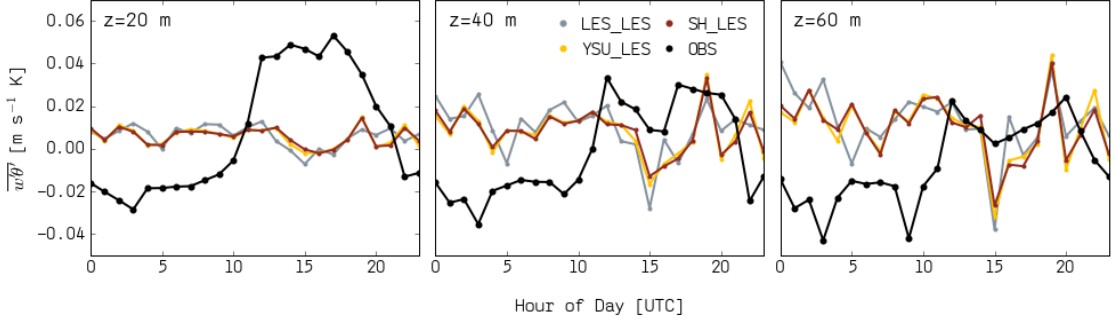

**Figure 10.** Mean diurnal cycle (local time is UTC-3 during experiment) of kinematic heat flux $\overline{w'\theta'}$ [m s$^{-1}$ K]. The mean is computed from all 10-minute periods for which simulations and measurements are available at each hour.

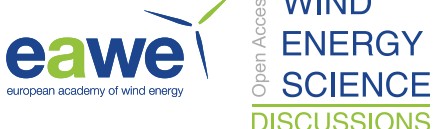

The normalized difference in simulated kinematic heat flux between `SH_LES` and `YSU_LES`

$$\frac{\overline{w'\theta'}_{\text{SH\_LES}}(t) - \overline{w'\theta'}_{\text{YSU\_LES}}(t)}{\frac{1}{N_t}\sum_{t=1}^{N_t}\overline{w'\theta'}_{\text{YSU\_LES}}(t)} \tag{1}$$

can be very large ($\mathcal{O}(10^3)\%$) at specific time steps but over the entire period it is $\sim 1.5\%$ at 20 m, $\sim 0.1\%$ at 40 m, and $\sim -0.1\%$ at 60 m. Note that in Shin and Hong (2013) a constant surface heat flux is used to examine the stability dependence

on grid-size. Here, we reveal that in a full-physics simulation where surface forcings are unsteady, the scale-awareness of this ABLP can have the effect of decreasing the heat flux simulated by the model under a variety of atmospheric conditions, even when a long-term averaged value is considered.

### 4.3 Energy Spectra

Fig. 11 shows the spectral energy content in the 10-minute mean wind speeds at each measurement height, computed using

the methods described in Section 2.5 and considering only the days in which the measurement data set was complete. Once again, the difference between `YSU_LES` and `SH_LES` can hardly be discerned. The spectral interval characterized by the frequencies $f \in \left(4 \times 10^{-5}, 5 \times 10^{-4}\right)$ Hz (time scales between $\sim 7$ hours and $\sim 30$ minutes) reveals a clear and consistent underestimation of the energy content by all of the simulations and at all three heights, clearly marking the terra incognita described in Section 1. The analysis for this period and site suggest that running the GZ domain with an ABLP produces better

results in terms of variance of the streamwise velocity than configuring this domain in LES mode. This analysis complements the discussion presented in Section 3.3 by demonstrating that the three simulations perform well below and above GZ scales, where the energy decay follows a $f^{-5/3}$ slope which was not evident from the model cross-comparison alone. While the observations show a $f^{-5/3}$ relationship throughout the frequencies considered, the simulations present a $f^{-1}$ slope at the GZ scales as previously discussed.



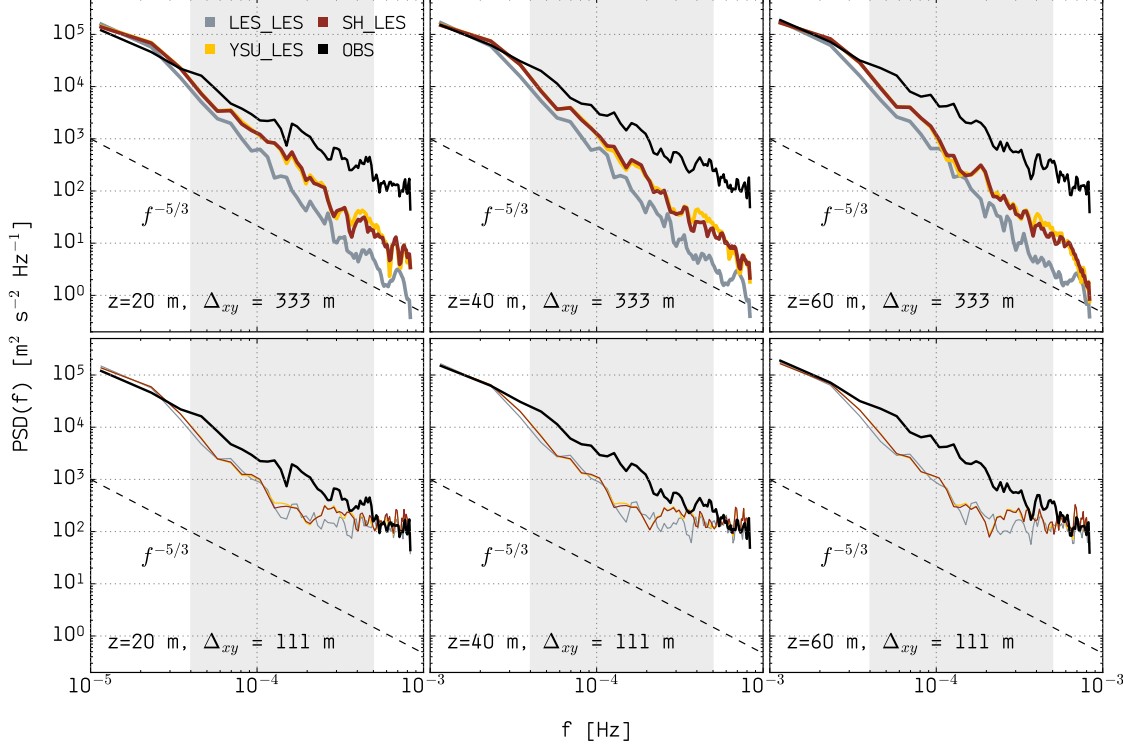

**Figure 11.** Power spectral density as a function of frequency for horizontal wind speeds at $z = 20$ m (left), 40 m (middle) and 60 m (right) for observations (black), `LES_LES` (gray), `YSU_LES` (blue), and `SH_LES` (red). Top shows modeled fields at GZ domain, and bottom at innermost domain. Spectra consider all 10-minute mean time stamps in which measurements and simulations were available and an entire 24-hour period was complete. Gray shading marks GZ spectral region. Top figures are for the GZ domain, and bottom figures for the micro-scale domain.

## 5   Conclusions

Due to ongoing advances in computational power, meso-micro scale coupling for wind energy applications has recently become a topic of active research. We are currently moving away from the need to choose between (i) idealized simulations at high spatial and temporal resolutions or (ii) real case simulations limited to the meso scale. In this study, we conduct multi-day, real

5   case, full-physics simulations of the atmosphere down to a spatial resolution of 111 m, and consider different ways of treating the terra incognita or GZ which characterizes the transition from meso to micro scales. At this transition, a large portion of the kinetic energy is naturally produced by the momentum balance equation in the model, while the remaining part still needs to be parameterized. We conducted three simulations that only differ in their treatment of the ABL in the GZ domain. One of them uses the traditional YSU parameterization, another uses its scale-aware version SH, and a final one uses no parameterization

10   at all and runs the GZ domain in LES mode. We seek to quantify how different GZ treatments affect flow simulations at





the GZ domain and at the micro scale domain nested within it, and to determine which simulation performs best relative to measurements.

We find that running the GZ domain in LES mode produces more fluctuations in the horizontal and vertical wind fields, and that it has a large impact on the simulated flow, with temporally averaged differences in the horizontal wind speed and
direction as large as 1 m s$^{-1}$ and 10° between LES_LES and an ensemble mean of the three simulations. The distribution of wind speed errors showed clear trends with atmospheric conditions, but the tendency was uniform across the three simulations. Differences in mean wind speed errors between the simulations are small, with SH_LES and YSU_LES performing slightly better for low turbulence intensity and for offshore flow than LES_LES.

The LES_LES simulation produced higher vertical velocities and TKE at the gray zone domain, which led to more homo-
geneous flows and lower levels of shear when compared to SH_LES and YSU_LES. The best shear predictions were those by the parameterized runs at the GZ domain, suggesting that the treatment of the ABL at GZ resolutions is more critical to accurate simulations of shear than increasing the spatial resolution with an extra nested domain. These results are likely to differ if feedback is enabled between the domains, which was not considered in the present study.

Differences between YSU_LES and SH_LES were found to be negligible when multi-day temporal averages are consid-
ered, and much lower in magnitude than the model error relative to measurements. However, differences between these two simulations are discernible over short intervals not only at the GZ domain but also at the inner nest, which uses data from the GZ domain as boundary conditions. These differences are most pronounced for kinematic heat flux, where SH_LES produces peaks of slightly lower magnitude than YSU_LES during daytime. None of the simulations capture the diurnal cycle of the heat flux near the surface, indicating that this systematic error might be related to surface layer physics instead of ABL physics.

The spectral analysis indicates that including a micro scale domain within the model chain leads to a substantial recovery in the decaying tails of the spectral energy content, and clearly reveals the existence of a GZ or terra incognita in the frequency range between 7 hours$^{-1}$ and 30 minutes$^{-1}$. The PSD computed for wind speeds simulated in the GZ domain follow a $f^{-3}$ slope throughout most of the spectral range considered. In the micro scale domain, the same is only seen in the low-frequency end of the GZ spectral range. In the high-frequency end, the energy decay follows a $f^{-1}$ slope. Outside the GZ spectral range,
the micro scale simulated spectra follow the expected $f^{-5/3}$ slope and produce good agreement with measurements. Overall, the parameterized simulations SH_LES and YSU_LES produce streamwise velocity variance values that more closely match the observations at both resolutions.

Overall, we found that for multi-day simulations the present question is whether to run the GZ in LES mode or with a ABLP at all, and not whether to consider scale-aware parameterizations which are still in early development phases. For the period
considered, very small differences were seen between YSU and its scale-aware version, SH. For shorter time periods, the differences between SH_LES and YSU_LES can be substantial and the GZ treatment should be considered more carefully. With the current data set, we cannot yet determine whether SH_LES could result in a substantial improvement of the simulated fields under non-idealized simulations. We generally found that parameterizing the GZ domain results in more accurate predictions of shear, TKE, and spectral energy content. Further work is needed to generalize the development of scale-aware parameter-



izations so that they may provide a solution for fully coupled, real case simulations of the ABL with realistic predictions for both shear and turbulence.



*Author contributions.* P. Doubrawa performed the analysis and wrote the manuscript. R. J. Barthelmie and S. C. Pryor obtained the funding and designed the field experiment, participated in observational data collection, contributed to discussions and thoroughly reviewed the manuscript. P. Casso performed the simulations. A. Montornès developed the WRF-LES framework for the simulations, participated in discussions about the data analysis and thoroughly revised the manuscript.

5   *Competing interests.* The authors declare that they have no conflict of interest.

*Acknowledgements.* This work was partly funded by National Science Foundation 1565505, and the Department of Energy DE-EE0005379 and DESC0016438. The measurements were performed at the Wind Energy Institute of Canada. The computational resources were provided by VORTEX.



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
