# Peer review of "Analysis of Different Gray Zone Treatments in WRF-LES Real Case Simulations"

_Wind Energy Science, 2017_

## Referee Comment (RC1) · Anonymous Referee #1 · 13 Feb 2018

In this article, the authors present a sensitivity analysis of the turbulence closure approach utilized in an intermediate domain with grid spacing within the gray zone (GZ) when performing coupled mesoscale-LES simulations with the WRF model. This is an interesting aspect of multiscale simulations that requires quantification, and the authors present systematic analysis to that end, including comparison to field observations from the Prince Edward Island Wind Energy Experiment. Overall the manuscript is well written and logically organized. However, there are major aspects that must be addressed before the manuscript can be considered for publication in Wind Energy Science. The two main issues are:

1) Figure 9 clearly exhibits that the model results are not able to capture the basic diurnal evolution of the atmospheric boundary layer (ABL), with the near-surface sensible

heat flux being nearly constant in time throughout the day in the model (∼0.005 K m s-1). This is strong evidence that there is something wrong with the model setup. The model does not capture the diurnal cycle and therefore, any attempt to see the impact of the GZ on the nested LES results through comparison to observations is hopeless. While a multi-day period as the authors present here is desirable, it does not add much if the results are not properly representing the meteorological conditions of interest to a minimum degree. The authors should look for cases within this 16-day time window (if any) where mesoscale results reasonably compare to the observations, at least in terms of basic features, and exclude the rest of the days from the analysis. If none is found, the authors need to explore the use of different re-analysis datasets and/or physics options to try to find a better model performance at the mesoscale before running the GZ and nested LES simulations.

2) While the grid resolution employed at the innermost LES domain may be suitable for daytime conditions; it is definitely not the case for the stable boundary layer. A horizontal grid spacing of 111 m is too large to resolve the characteristic turbulent scales that dominate stable boundary layers, typically of the order of few meters to several tens of meters. This implies that resolutions of at most 10 m are required, as it is often used in both ideal and real world LES modeling of stable ABLs. With that consideration in mind, the authors should either change their setup consistently with this requirement or to remove stable ABL instances from the analysis.

Here a list of other comments, following the order of appearance in the manuscript:

- Page 1, line 2: "momentum". Is not only momentum balance, also energy equation is involved. Please modify to: "produced by the model".

- Page 1, line 7: It would be more correct to say an LES parameterization. Or, a "large-eddy simulation (LES) parameterization where the most energetic turbulent motions are resolved and only the effects of the sub-grid scales on the resolved flow are parameterized".

- Page 1, line 13: Any explanation for the -3 slope?

- Page 1, line 21: "Navier-Stokes equations".

- Page 2, line 7: "fluid dynamcis". You should employ a better term. NWP models are "geophysical fluid dynamics" models. You could perhaps use instead "engineering computational fluid dynamics (CFD)" models, which is more commonly used in that context.

- Page 2, lines 10-13: This literature review is incomplete. There are other works in the literature including these by Munoz-Esparza et al. JAMES2017 and Rai et al. BLM2017 that need to be acknowledged here.

- Page 2, line 19: "LES mode". Please add some references here.

- Page 2, line 20: "single code". Please add some references here.

- Page 2, line 26: "GZ is avoided". Better would be to say: "can potentially be avoided". However, the problem remains, since one likely wants to use high-resolution mesoscale forcing to drive the stand alone LES calculations. It could also be avoided within the same model by skipping GZ resolutions.

- Page 3, line 1: "Mirocha et al. (2014)".

- Page 3, line 25-27. It is unclear what this has to do with GZ. Please explain.

- Page 4, line 2: "no ABLP at all". You can perhaps name that very large-eddy simulation (VLES) to avoid confusion with the LES in the nested domains. - Page 4, line 20. The appropriate reference is apparently missing in the references section. "Bridging the transition...." Boundary-Layer Meteorology (2014) 153:409-440.

- Page 5, line 3-4: Are you using SH for stable boundary layers as well? If so, you should mention that SH parameterization was developed for convective ABLs, and that is use for other stabilities remains questionable...

- Page 5, line 7-10: This speculation may not be needed here, since this is what you will be analyzing in detail in the remaining of the manuscript.

- Page 5, line 11: "Study Domain". What do you mean? NWP model domain? This is confusing, please change.

- Page 5, line 23: "indicates".

- Page 5, line 24: Is it neutral the rest of the time? What is the threshold in L used to determine the classification? I guess it is described in the reference, but it would be a good help to the reader if it is mentioned here as well.

- Page 6, lines 6-7: How can that be possible? What is the top pressure? A CBL can easily be several km deep, in which case very few grid points remain for the rest of the atmosphere. Could you instead provide the \Delta z of the first and last levels within the ABL.

- Page 6, line 10: Please include the rest of model physics options used in the study.

- Page 6, line 11: What about spinup time? Did you perform multiple initializations or not? Please describe. If a single run, is the SST updated? Also, make sure the landuse is such that the gird cell where you are outputting the results is considered as "land". If that is not the case, such mismatching would explain the issues with the diurnal cycle...

- Page 7, line 8: "including the ABLP budget". What does that mean? Which budget?

- Page 7, line 9: It would be interesting to see the sub-grid scale contribution (and total) for the GZ domain as well.

- Page 8, line 1: Results in this section (Figs. 3, 4, 5, 7, 8, 9) should be separated at least into two stability classes (convective and stable). The model errors can cancel out and its nature is expected to be stability-dependent. Otherwise, it is difficult to identify any clear trends in the analysis. In particular, stable results are highly questionable (as

mentioned earlier), and therefore should be separated from the rest not to corrupt the rest of the analysis.

- Page 10, line 2-4: It is somewhat puzzling that although the authors claim the LES_LES is the least performing, the PDF in the GZ LES is more similar in structure to all the microscale LES (although the peak is over-predicted). So the micro LES departures from the GZ ABLP solutions. Could the authors ellaborate on this?

- Page 10, line 5: There is no budget presented, just TKE. Remove "budget".

- Page 11, line 1: Figure 5. Why do you use U to correlate with TKE? It would be more appropriate to plot time of the day, since that is likely a better estimate of stability than U.

- Page 11, line 3-5: This is not correct. The PSD you are seeing at low frequencies is that of mesoscale motions, which are essentially quasi-two-dimensional in nature. The ABLPs act as diffusion terms, and will therefore extract energy from the model. Also, how can there be more variability in the GZ LES TKE from Fig. 5 and less energy in the PSD? "Act to generate turbulence energy at length scales higher than the grid size". How is that possible? What is the mechanism? Please explain.

- Page 11, line 10-11: WRF has been shown to produce mesoscale -5/3 kinetic energy spectra (e.g., Skamarock 2004). Please plot the spectrum from the 3 mesoscale domains and see whether that pattern is induced by the parent domains or not.

- Page 11, line 14-16: In the paragraph just above you claim usage of the 4 Hz output in the previous section...

- Page 13, Figure 7: Any explanation of why the wind speed bias is very similar between all the GZ-driven simulations while there were large differences in between the GZ solutions?

- Page 13, line 10-13: The differences here are smaller than for other wind directions. That is a plausible explanation but you need to support that with additional analysis.

- Page 14, line 7: The micro LES distribution of the LES_LES case is quite similar in structure to the ABLP-driven cases, clearly departing from the 333 m distribution. This points to some issue/s in the modeling. Could the authors please comment on this?

- Page 14, line 14: This is hard to understand. If errors were larger for low wind speeds, what is causing the best TKE agreement?

- Page 16, Sect. 4.3: Given that there is no diurnal-cycle forcing in the simulations, and such is the extent of the frequency range considered in this section, this analysis is totally inconclusive (it compares to very different things, model and observations). Needs to be repeated once a better model setup is found.

---

## Referee Comment (RC2) · Anonymous Referee #2 · 15 Feb 2018

Summary and Overall Recommendation:

This article presents a systematic comparison of the effect of different gray zone (GZ) turbulence closure treatments, as well as their effect on a microscale nest. In doing so, the authors address an important problem that arises when coupling between mesoscale and microscale models. The organization of the paper is logical and easy to follow, with comparisons between the GZ models followed by comparisons between the models and field data from the Prince Edward Island Field Energy Experiment. However, major revisions are needed before this work can be published in Wind Energy Science. In particular, simulations show be reran using finer resolutions for the finest domain, and the presence of GZ effects in the middle domain should be demonstrated.

Major Concerns:

1. The lack of significant differences between the YSU and SH results points to a potential issue with the modeling setup. It is possible that the selection of the SH scheme was not appropriate to the modeled conditions (Please, justify selection of this scheme despite its design being limited to convective conditions - see Specific Comment #7), or that the resolution selected for the middle domain did not present any gray-zone artifacts or biases. Were any preliminary simulations done to ensure that the grid-cell size of 333m was within the gray-zone for all of the atmospheric conditions presented, and presented issues when being modelled through the YSU PBL scheme? If a resolution of 333m is not resolving turbulence within the inertial range, it is possible that the GZ modifications of the SH scheme are not being used, thus explaining the lack of differences between YSU and SH, but also indicating that the center domain does not actually represent the GZ. If any tests were done to justify the resolution selection of the GZ, please include those. 2. A finest resolution of 111m seems like a coarse value for certain atmospheric conditions. Were tests performed to confirm that it can appropriately represent stable turbulent motions, which are characterized by smaller scales? If this resolution is not fine enough to resolve the inertial range of turbulence within the nested LES domains, it is possible that GZ issues are present in this domain.

Specific Comments:

1. Page 1, Line 22 - That LES resolves "all of the turbulence relevant to the problem at hand" is an extreme assumption.

2. Page 2, Table 1 - Replace "<" with ">" for the Macro and Meso Scale cases.

3. Page 2, Lines 17 to 20 - The words "coupled" and "uncoupled" are incorrectly used to describe one-way and two-way nesting instead of coupled and uncoupled models.

4. Page 2, Lines 21 to 22 - The second question posed here ("how best to treat the transition from meso to micro sacles within numerical models") is very vague. Is the author referring to "how best to treat the GZ within coupled numerical models"?

5. Page 2, Lines 26 to 27 - Is the "first coupling approach" the one that was previously referred to as an "uncoupled" approach? If so, this is another argument to change the use of the words "coupled" and "uncoupled" in lines 17 to 20.

6. Page 2, Lines 26 to 27 & Page 3, Lines 1 to 4 - Please explain how this approach avoids the GZ. Wouldn't it also benefit from higher-resolution, properly treated GZ domain boundary conditions? Similarly, the second approach in this section states that an adequate treatment of the GZ is "required", yet later on it is stated that the GZ can be avoided altogether, thus not requiring this treatment. The definition of, and differences between these two approaches are not clear.

7. Page 3, Line 11 - It is stated that the SH scheme is limited to convective boundary layers. However, in this study it was selected as a modeling tool for several complete diurnal cycles, including non-convective atmospheric conditions. Please, explain how this selection is justified (see Major Comment #1).

8. Page 6, Section 2.3 - It was previously stated that full-physics were used for this study. What physics schemes were used for this? Please provide more details. Also, how were the simulations spun-up?

9. Page 6, Line 4 - Please, specify that '9 km to 111 m' refers to horizontal grid-cell size.

10. Page 7, Section 2.5 - It is unclear why the spectra was calculated using 10-minute mean winds rather than the 4 Hz model output data for the intra-simulation comparisons.

11. Page 7, Line 26 - "sufficient when focusing on GZ phenomena": Please, explain/justify or cite sources.

12. Page 9, Line 3 - Since the YSU and SH models are very closely related, it can be expected that their results are very similar. This would be especially true if the chosen GZ resolution did not, actually, correspond to the GZ for the particular configuration

and atmospheric conditions (SEE Major Comment #1). Therefore, rather than stating that the LES mode has a large impact on the results, a more accurate statement would be that "the choice of turbulence modeling scheme has a large impact in the results"

13. Page 10, Lines 5 to 10 - What is TKE budget referring to? Should this just represent total TKE, as described in Page 7, Line 7?

14. Page 10, Line 8 - YSU and SH should still contain parametrized turbulence, which must be taken into account when computing total TKE. Please, check that this has been included in the total TKE, plotted in Figure 5.

15. Page 10, Line 9 - It is not obvious that TKE would be the same for all three simulations, since the SGS or parametrized TKE does, indeed, have an effect on the resolved flow. 16. Page 10, Line 13 - Please, describe "these differences". It is not obvious that they are similar to those seen in the GZ domain.

17. Page 10, Line 15 - Where is TKESGS being shown? Figure 5 (which we understand to show total TKE) shows an increase up to U around 14 m s-1, followed by a sharp decrease.

18. Page 11, Figure 5 - This figure shows nearly no turbulence for the YSU or SH schemes. From the paper it seems like the value of TKE being plotted is total (SGS + resolved). However, this figure seems more representative of resolved TKE values.

19. Page 11, Figure 5 (caption) - modify "TKE at 60 m [m2 s-2]" to "TKE [m2 s-2] at 60 m"

20. Page 11, Lines 3 to 5 - Please, elaborate. This result doesn't seem very obvious. How does the ABLP generate turbulence at length scales higher than the grid-size? This sentence is not very clear.

21. Page 11, Lines 14 to 15 - Is this data not available from the 4Hz model output used to calculate TKE?

22. Page 12, Figure 6 - Section 2.5 states that the spectra are computed for 1 day-1 < f < 20 min-1. However, the range of frequencies being plotted in this figure only go up to 10-3 Hz, which does not correspond to 20 min-1. Could this be clarified?

23. Page 12, Line 13 - Could the fact that the "YSU_LES and SH_LES error distribution is almost indistinguishable" be an indication of an issue with the setup resolutions? (See Major comment #1)

24. Page 14, Line 4 - The good performance of the YSU and SH schemes could be related to the choice of resolutions. As mentioned in Major Comments #1 and #2, if the resolution of the middle domain is not indeed within the GZ, and the resolution for the finest LES domain is too coarse, the observed results may be explained. 25. Page 14, Lines 11 to 13 - If the smallest domain resolution is too coarse for LES modeling (which is possible, since such a resolution may be too coarse for the stable conditions that were commonly observed during the experiment), then the finest domain cannot be assumed to be performed at an LES scale, and the conclusion drawn on this sentence would not be accurate.

26. Page 15, Figure 10 - This type of plot, showing the evolution of a turbulence quantity in time, could be more insightful for analyzing TKE, wind speeds and wind speed errors than those with respect to U (Figures 2, 5 and 9) or other flow quantities (figure 7).

27. Page 16, Lines 13 and 14 - The energy spectra is being computed from resolved turbulence. Therefore, it makes sense that all three simulations underestimate the energy that is measured, since some of this energy is in the sub-grid scales. Please, elaborate as to how this lower turbulence explains the presence of a GZ or correctly incorporate the SGS TKE.

28. Page 16, Section 4.3 - The analysis in this and other sections would benefit from being separated by atmospheric stability conditions, since stability conditions may have a large influence on the results.

29. Page 17, Lines 9 to 10 - Throughout the article, LES simulations are treated as not-parameterized, while the YSU and SH simulations are considered to be parameterized. This statement is used to draw conclusions in some of the analysis (i.e. specific comment #20). However, a turbulence closure scheme is still used for LES, and sub-grid turbulence is still calculated and affects the resolved flow. Therefore, LES is still a type of parameterization, albeit different from the Planetary Boundary Layer (PBL) schemes. Some clarification about what is meant by parameterized and un-parameterized, or the use of more precise and specific language, would be helpful.

30. Page 18, Lines 11 to 12 - As mentioned in specific comment #25, if the finest domain does not have high enough resolution, this conclusion about the benefits of adding an extra nested domain cannot be drawn without considering the possible effect of the GZ on this finest domain.

31. Page 18, Lines 18 to 19 - That none of the simulations are able to capture the diurnal cycle could also be an indication that the current setup may not be correct for the questions that it is trying to answer.

32. Page 18, Line 21 - It is not clear how the presence of the GZ is confirmed by this (see specific comment #27).